# Engineering of CD19 Antibodies: A CD19-TRAIL Fusion Construct Specifically Induces Apoptosis in B-Cell Precursor Acute Lymphoblastic Leukemia (BCP-ALL) Cells In Vivo

**DOI:** 10.3390/jcm10122634

**Published:** 2021-06-15

**Authors:** Dorothee Winterberg, Lennart Lenk, Maren Oßwald, Fotini Vogiatzi, Carina Lynn Gehlert, Fabian-Simon Frielitz, Katja Klausz, Thies Rösner, Thomas Valerius, Anna Trauzold, Matthias Peipp, Christian Kellner, Denis Martin Schewe

**Affiliations:** 1ALL-BFM Study Group, Department of Pediatrics I, Christian-Albrechts University Kiel and University Medical Center Schleswig-Holstein, Arnold-Heller Str. 3, Haus C, 24105 Kiel, Germany; dorothee.winterberg@uksh.de (D.W.); Lennart.lenk@uksh.de (L.L.); Fotini.Vogiatzi@uksh.de (F.V.); 2Department of Medicine II Christian-Albrechts, Division of Stem Cell Transplantation and Immunotherapy, Campus Kiel, University Kiel and University Hospital Schleswig-Holstein, 24105 Kiel, Germany; osswaldmaren@gmail.com (M.O.); CarinaLynn.Gehlert@uksh.de (C.L.G.); Katja.Klausz@uksh.de (K.K.); Thies.Roesner@uksh.de (T.R.); t.valerius@med2.uni-kiel.de (T.V.); m.peipp@med2.uni-kiel.de (M.P.); 3Institute for Social Medicine and Epidemiology, University of Lübeck, 23538 Lübeck, Germany; fabian@frielitz.net; 4Institute for Experimental Cancer Research, Christian-Albrechts-University Kiel, 24104 Kiel, Germany; atrauzold@email.uni-kiel.de; 5Department of Transfusion Medicine, Cell Therapeutics and Hemostaseology, LMU University Hospital Munich, 81377 Munich, Germany; Christian.Kellner@med.uni-muenchen.de

**Keywords:** BCP-ALL, leukemia, TRAIL, antibody, Fc-engineering, xenograft, CD19

## Abstract

B-cell precursor acute lymphoblastic leukemia (BCP-ALL) is the most frequent malignancy in children and also occurs in adulthood. Despite high cure rates, BCP-ALL chemotherapy can be highly toxic. This type of toxicity can most likely be reduced by antibody-based immunotherapy targeting the CD19 antigen which is commonly expressed on BCP-ALL cells. In this study, we generated a novel Fc-engineered CD19-targeting IgG1 antibody fused to a single chain tumor necrosis factor (TNF)-related apoptosis-inducing ligand (TRAIL) domain (CD19-TRAIL). As TRAIL induces apoptosis in tumor cells but not in healthy cells, we hypothesized that CD19-TRAIL would show efficient killing of BCP-ALL cells. CD19-TRAIL showed selective binding capacity and pronounced apoptosis induction in CD19-positive (CD19^+^) BCP-ALL cell lines in vitro and in vivo. Additionally, CD19-TRAIL significantly prolonged survival of mice transplanted with BCP-ALL patient-derived xenograft (PDX) cells of different cytogenetic backgrounds. Moreover, simultaneous treatment with CD19-TRAIL and Venetoclax (VTX), an inhibitor of the anti-apoptotic protein BCL-2, promoted synergistic apoptosis induction in CD19^+^ BCP-ALL cells in vitro and prolonged survival of NSG-mice bearing the BCP-ALL cell line REH. Therefore, IgG1-based CD19-TRAIL fusion proteins represent a new potential immunotherapeutic agent against BCP-ALL.

## 1. Introduction

B-cell precursor acute lymphoblastic leukemia (BCP-ALL) is the most frequent childhood malignancy. Whereas most patients can be cured by chemotherapy, this is associated with severe side effects, and relapse remains a major clinical challenge [1,2,3]. Immunotherapeutic approaches, especially monoclonal antibodies, exert highly specific anti-tumoral efficacy with lower off-target toxic effects [4]. Accordingly, antibody-based immunotherapy is being introduced into the treatment of B-cell malignancies including BCP-ALL, both in frontline therapy and in the treatment of relapsed and refractory disease [5,6]. An attractive therapeutic target in BCP-ALL is the pan B-lymphocyte antigen CD19, a type I membrane protein of the immunoglobin superfamily that is expressed by the majority of B-lineage lymphoid malignancies [7,8,9,10]. To this end, targeting CD19 with novel immunotherapeutic approaches, such as the (CD3 × CD19) bispecific T-cell engager molecule (BiTE) blinatumomab or chimeric antigen receptor (CAR) T-cells, have entered routine clinical care in specific situations [11,12,13]. CD19 antibody-drug conjugates (ADC) such as coltuximab ravtansine (SAR3419) have shown tolerability but poor clinical response in patients with relapsed or refractory BCP-ALL [14]. Native CD19-IgG1 antibodies displayed only limited efficacy in preclinical models [15,16]. Yet, the therapeutic efficacy of CD19 antibodies can be improved by fragment crystallizable (Fc)-engineering, e.g., through introducing amino acid substitutions into the heavy chain (CH) region 2 or by changing the glycosylation pattern of the antibody. As a result, the affinity to Fcγ receptors on effector cells is increased, leading to enhanced effector cell recruitment and activation [15,17,18]. We previously showed that an Fc-engineered CD19 antibody carrying a S239D/I332E mutation (DE-modification) showed enhanced effector cell-mediated killing of tumor cells and pronounced efficacy in BCP-ALL xenografts in vivo [19]. The DE-modified antibody tafasitamab is currently being tested in BCP-ALL patients (ClinicalTrials.gov identifier NCT01685021).

Another promising antibody modification is the linkage with biological cytotoxic agents such as the tumor necrosis factor (TNF)-related apoptosis-inducing ligand (TRAIL) [20]. TRAIL is a homotrimeric type II transmembrane protein that initiates extrinsic apoptosis by binding its agonistic death receptors TRAIL-Receptor 1 (TRAIL-R1) and TRAIL-R2 on the target cell [21,22,23,24,25]. This results in receptor oligomerization and subsequent assembly of the death-inducing signaling complex (DISC) and activation of a caspase cascade [26]. Of note, TRAIL was shown to induce apoptosis in cancer cells selectively, even in the absence of a high proliferation rate [25,27,28,29]. Treatment with recombinant TRAIL showed promising results in preclinical studies [30,31]. However, clinical pilot studies with non-small-cell lung cancer and relapsed follicular non-Hodgkin’s lymphoma patients found no superior outcomes when adding TRAIL to standard care [32,33]. Proposed reasons for the limited in vivo efficacy are the instability and rapid clearance of TRAIL as well as the apoptosis resistance of tumor cells [31,34]. The latter may be based on the presence of TRAIL-decoy receptors and the widespread TRAIL-R expression in the tumor microenvironment competing for TRAIL ligands, thereby limiting the accumulation of TRAIL on tumor cells [21,35,36]. These limitations can be overcome by fusing TRAIL to tumor-specific antibodies or antibody-fragments, particularly constructs based on IgG structures (IgG-like), which generally harbor a superior pharmacokinetic profile [20,37,38]. These fusion constructs may accumulate on the pre-selected target antigen of tumor cells and lead to the subsequent anchoring of the TRAIL domain on the cell surface promoting increased TRAIL-R engagement. Such constructs may even outperform Fc-less fusion proteins [39,40]. Therefore, we hypothesized that genetic fusion of a monoclonal CD19-directed IgG antibody to monovalent single-chain (sc) TRAIL generates a fusion protein that combines the specificity and beneficial pharmacokinetics of an IgG antibody with the cytotoxic activity of a tumor-cell-specific death ligand.

Here, we report the successful generation of such an IgG fusion protein (CD19-TRAIL) and show that CD19-TRAIL efficiently kills CD19-positive (CD19^+^) BCP-ALL cell lines in vitro and in vivo and is also effective in BCP-ALL xenograft mouse models. Moreover, we show that the tumor-killing effect of CD19-TRAIL can be synergistically enhanced by dual induction of the extrinsic and intrinsic apoptosis pathways. Our preclinical data suggest that CD19-IgG-antibodies coupled to scTRAIL may be an effective agent to target BCP-ALL cells directly.

## 2. Materials and Methods

### 2.1. Cell Culture

The BCP-ALL cell lines REH and NALM-6 and the T-ALL cell line CEM were obtained from the German Collection of Microorganisms and Cell Cultures (DSMZ, Braunschweig, Germany) and cultured in RPMI 1640 Glutamax-I medium containing 10% fetal calf serum (FCS), 100 U/mL penicillin, and 100 µg/mL streptomycin (Thermo Fisher Scientific, Waltham, MA, USA). Chinese hamster ovary (CHO-S) cells were purchased from Thermo Fisher Scientific and cultured in an orbital shaker with serum-free CD-CHO medium containing 1% HT supplement and 2 mM GlutaMax (Thermo Fisher Scientific, Waltham, MA, USA). For culturing of transfected CHO-S cells, the CD OptiCHO medium was supplemented with 1% HT supplement, 2 mM GlutaMax, and 0.1% Pluronic F-68 (Thermo Fisher Scientific, Waltham, MA, USA).

### 2.2. Antibodies

CD19-TRAIL and HER2-TRAIL were generated by de novo synthesis of the DNA sequences of the variable regions of the antibodies tafasitamab (MOR208) and trastuzumab, respectively, according to published sequences [41,42]. To promote heterodimeric Fc domain pairing, codon exchanges for amino acid substitutions in the CH3 domain (i.e., K392D and K409D for the first HC and E356K and D399K for the second HC) were respectively inserted [43], and a DNA sequence encoding a single chain TRAIL domain containing three TRAIL monomers [44] was genetically fused to the C-terminal part of the first HC.

Sequences were ligated in frame into antibody light chain (LC) expression vector pSECtag2-LC or antibody HC expression vector pSECtag2-HC [17,19]. Validity of the cloned sequences was confirmed by Sanger sequencing, and plasmid DNA was purified with Nucleo Bond 2000 EF (Macherey-Nagel, Düren, Germany). Antibody-TRAIL fusion proteins and a native CD19-IgG1 antibody [19] were produced in CHO-S cells by transient transfection of HC and LC expression vectors using the MaxCyte STX Scalable Transfection System (MaxCyte, Gaithersburg, MD, USA), as previously described [45]. Antibodies were purified from cell culture supernatant with a CaptureSelectTM IgG-CH1 affinity matrix (Thermo Fisher Scientific, Waltham, MA, USA) and size exclusion chromatography (ÄKTA Pure 25 liquid chromatography system; GE Healthcare, Chicago, IL, USA).

### 2.3. Sodium Dodecyl Sulfate Polyacrylamide Gel Electrophoresis (SDS-PAGE) and Western Blot Analysis

Antibody integrity and concentration were analyzed by SDS-PAGE under reducing conditions and western transfer experiments, as published previously [46].

For detection of human IgG HC, LC, and TRAIL, a goat-anti-human-IgG-HRP conjugate (#AP113P, Sigma Aldrich, St. Louis, MO, USA), a mouse anti-human kappa light chain (#K4377, Sigma Aldrich, St. Louis, MO, USA), and a rabbit anti-human TRAIL antibody (#ab9959, Abcam, Cambridge, UK) were used, respectively.

### 2.4. CD19 Binding Capacity

Specific binding capacities of CD19-TRAIL, CD19-IgG1, and HER2-TRAIL were analyzed as previously reported using a secondary anti-human IgG Fc F(ab′)2 polyclonal goat antibody conjugated to Fluorescein isothiocyanate (FITC) (Jackson Immuno Research, West Grove, PA, USA) [47]. Flow cytometric analyses were performed with a Navios flow cytometer, and data were analyzed with the Kaluza 1.2 software (Beckman Coulter, Brea, CA, USA).

### 2.5. Expression of CD19, TRAIL-R1, and TRAIL-R2 on Cell Lines and PDX Samples

Quantification of expression levels of CD19, TRAIL-R1, and TRAIL-R2 on the cell surface was analyzed by flow cytometry using QIFIKIT (Agilent Technologies, Santa Clara, CA, USA) according to the manufacturer´s protocol. Briefly, cells were washed with PBA (phosphate-buffered saline (PBS, Thermo Fisher Scientific, Waltham, MA, USA), 1% bovine serum albumin (BSA, Carl Roth, Karlsruhe, Germany), and 0.1% sodium-azide (Merck, Darmstadt, Germany). Then, 0.5 × 10^6^ cells were incubated with saturated concentrations of mouse antibodies against CD19 (#392502, Biolegend, San Diego, CA, USA), TRAIL-R1 (#307202, Biolegend, San Diego, CA, USA), and TRAIL-R2 (#307302, Biolegend, San Diego, CA, USA) for 1 h, respectively. After washing with PBA, cells were incubated for 30 min with a FITC-labelled anti-mouse antibody. Fluorescence was measured by flow cytometry using a MACSQuant X Analyzer (Miltenyi Biotec, Bergisch Gladbach, Germany). Antigen density was assessed by generating a standard curve obtained by beads coated with defined amounts of mouse IgG. Data were analyzed by FlowJo software Version 10.7.1 (Becton, Dickinson and Company, Ashland, OR, USA).

### 2.6. Cell Viability Assay

Direct cytotoxic effects were analyzed by 3-(4,5-Dimethylthiazol-2-yl)2,5-diphenyl tetrazolium bromide (MTT) assay (Roche Diagnostics, Mannheim, Germany). Briefly, 2 × 10^4^ cells per well in 100 µL medium were seeded in flat-bottom 96-well culture plates and treated with serial dilutions of indicated compounds for 72 h. After 4 h incubation with MTT reagent, the assay-solution was added to each well, and absorption at 550 nm (reference 650 nm) was measured after overnight culture. Cell viability was quantified as the percentage of growth inhibition compared to medium control (% relative growth of control). All experimental points were set up in triplicates.

### 2.7. Analysis of Apoptosis Induction

For analysis of apoptosis induction, cells were treated with 0.5 nM CD19-TRAIL and stained using AnnexinV-APC/PI (Biolegend, San Diego, CA, USA), and early and late apoptotic/necrotic cells were detected by fluorescence measurement using a Navios flow cytometer (Beckman Coulter, Brea, CA, USA). For apoptosis-blocking experiments, cells were pre-incubated with 50 µM of the pan-caspase inhibitor Z-VAD (MedChemExpress, Monmouth Junction, NJ, USA) prior to antibody incubation for 1 h. DMSO-treated cells served as a control. Data were analyzed with the Kaluza 1.2 software (Beckman Coulter, Brea, CA, USA).

### 2.8. Analysis of Drug Combination Effects

For analysis of the combined treatment effects of CD19-TRAIL with Venetoclax (VTX), 2 × 10^4^ cells were treated with CD19-TRAIL or VTX (LC Laboratories, Woburn, MA, USA) according to the determined EC50 value of either substance (0.06 nM and 6 nM, respectively) as well as increased and decreased concentrations in 5-fold dilution steps and then subjected to MTT assays. Drug synergy was assessed by calculation of the combination index (CI) using CompuSyn 1.0 software (ComboSyn, Inc., Paramus, NJ, USA) after 72 h [48]. Synergism, antagonism, or summation are indicated by CI < 1, CI > 1, or CI = 1, respectively [49]. For western blot analyses, cells were lysed after 48 h and analyzed using an apoptosis antibody sampler kit (#9915, Cell Signaling Technology, Danvers, MA, USA). Tubulin (#ab18251, Abcam, Cambridge, UK) was detected as a loading control.

### 2.9. Animal Experiments

Xenograft experiments were performed in accordance with governmental regulations (Schleswig-Holstein Ministerium für Energiewende, Landwirtschaft, Umwelt, Natur und Digitalisierung) using NOD.Cg-Prkdcscid Il2rgtm1Wjl/SzJ (NSG) mice bred in our institution.

For these experiments, 0.5 × 10^6^ REH, NALM-6, or 1 × 10^5^ BCP-ALL PDX cells were injected intravenously into 6–10 weeks old female NSG mice (day 0). CD19-TRAIL (1.5 mg/kg) was injected intravenously on day +1, +3, +6, +10, +13 as described previously [19] and every 7 days thereafter. Venetoclax (100 mg/kg) was applied daily by oral gavage [50]. Leukemic engraftment was analyzed via flow cytometric detection of human CD45^+^/murine CD45^−^/human CD19^+^ cells in the peripheral blood, and animals were sacrificed upon detection of >75% leukemic blasts or when showing clinical signs of leukemia (loss of weight or activity, organomegaly, hind-limb paralysis) as published previously [51,52,53].

### 2.10. Statistical Analysis

Graphical and statistical analyses were performed using software GraphPad Prism 9.0 (GraphPad, San Diego, CA, USA). If not stated otherwise, *p*-values were calculated using the Mann–Whitney test and repeated measures ANOVA with Bonferroni post-tests. For survival analyses, Kaplan-Meier and log-rank statistics were used. Significance was assumed when *p* < 0.05.

## 3. Results

### 3.1. Generation of a CD19-TRAIL Fusion Construct with CD19 Specific Binding Capacity

In order to generate a CD19-TRAIL fusion construct, a single chain TRAIL domain was genetically fused to the 3′-end of one HC of a CD19-IgG1 antibody [17]. As TRAIL-mediated activation of apoptosis relies on the trimerization of death receptors on the target cell surface [26], the TRAIL domain was designed to consist of three TRAIL units as previously reported [44]. To ensure heterodimeric HC pairing, amino acid exchanges were inserted in the CH3 domain: K392D and K409D for the first HC with the TRAIL domain and E356K and D399K for the second HC (Figure 1A). CD19-TRAIL was produced in CHO-S cells and purified by affinity chromatography. Aggregates and Fc homodimers were removed by size exclusion chromatography, resulting in a homogenous protein preparation with a single protein peak with a higher molecular mass compared to CD19-IgG1, indicating the insertion of the TRAIL unit (Figure 1B). SDS-PAGE under reducing conditions followed by Coomassie blue staining of purified CD19-TRAIL revealed three single bands according to the predicted molecular masses of the three different antibody-chains (LC = 25 kDa; HC = 50 kDa; HC + TRAIL = 112 kDa) (Figure 1C). Western blot analysis using a TRAIL-directed antibody further confirmed the successful incorporation of TRAIL into the antibody construct (Figure 1D). Next, CD19-TRAIL was tested for binding specificity to the CD19 antigen. CD19-TRAIL exposed equal binding capacity to the CD19^+^ BCP-ALL cell lines REH and NALM-6 as compared to the parental CD19-IgG1 antibody, while a HER2-TRAIL control antibody only showed significant binding due to TRAIL-R expression on the HER2-negative/CD19^+^ tumor cells (Figure 1E, Appendix A). Concordantly, no binding to CD19^−^ T-ALL CEM cells was observed (Figure 1E). Taken together, these data indicate the successful generation of a fusion construct of CD19-IgG1 and the death receptor ligand TRAIL with a specific binding capacity to CD19.

### 3.2. CD19-TRAIL Induces Direct Apoptotic Effects in CD19^+^ BCP-ALL Cells

Next, we characterized the CD19-TRAIL fusion construct with respect to target antigen-specific direct anti-leukemic efficacy. We first confirmed cell surface expression of CD19, TRAIL-R1, and TRAIL-R2 on the BCP-ALL cell lines REH and NALM-6 as compared to the T-ALL cell line CEM by quantitative indirect immunofluorescence analyses. As expected, the CD19 antigen was only expressed on REH and NALM-6 cells (mean specific antibody binding capacity (SABC) = 21,312 and SABC = 27,308, respectively), but not on CEM cells (SABC = 0, Appendix A). Furthermore, cell surface expression of TRAIL-R1 was low or absent (SABC: REH = 65; NALM-6 = 0; CEM = 41), in contrast to a strong TRAIL-R2 expression on all cell lines (SABC: REH = 503; NALM-6 = 2256; CEM = 4720) (Appendix A). We then examined the effect of the antibody on the viability of CD19^+^ REH and NALM-6 cells and CD19^−^ CEM cells by MTT assay. Indeed, CD19-TRAIL significantly reduced the viability of REH and NALM-6 cells at low nanomolar concentrations (EC_50_ = 0.0617 nM and EC_50_= 0.01746 nM, respectively) as compared to a HER2-TRAIL control antibody (Figure 2A,B). In line with higher CD19 and TRAIL-R2 levels on the cell surface, NALM-6 cells showed a response to CD19-TRAIL at lower concentrations as compared to REH cells (Figure 2A,B). As expected, the viability of CEM cells was neither affected by CD19-TRAIL nor HER2-TRAIL (Figure 2C). Of note, pre-incubation of CD19^+^ target cells with the parental CD19-IgG1 antibody, impeding binding of CD19-TRAIL, rescued the viability of REH and NALM-6 cells, further confirming target-specific effects of CD19-TRAIL (Figure 2D).

We next investigated whether CD19-TRAIL induces apoptosis in BCP-ALL cells. First, NALM-6 and REH cells were treated with CD19-TRAIL for 24 h, and exposure of phosphatidylserine on the outer cell membrane was assessed by AnnexinV staining at different time points. To distinguish early from late apoptotic or necrotic cells, co-staining with PI was performed. As expected, the progressive increase in AnnexinV-positive/PI-negative REH and NALM-6 cells showed apoptosis induction by CD19-TRAIL compared to untreated control cells (Appendix A). To confirm further apoptosis induction as a mechanism of killing, REH and NALM-6 cells were treated with the pan-caspase inhibitor Z-VAD prior to CD19-TRAIL treatment. Indeed, AnnexinV/PI staining revealed a significant reduction in cell death compared to CD19-TRAIL-treated cells (*p* = 0.05, respectively, Figure 3A,B). To confirm further CD19-TRAIL-mediated apoptosis induction, CD19^+^ REH and NALM-6 cells as well as CD19^−^ CEM cells were treated with CD19-TRAIL, CD19-IgG1, HER2-TRAIL, or a vehicle control, and cell lysates were analyzed for apoptosis-related proteins by western blot. As expected, we observed cleavage of the apoptosis markers Caspases 3, 7, and 9 as well as Poly (ADP-ribose) polymerase (PARP) in REH and NALM-6 cells in response to CD19-TRAIL treatment as compared to CD19-IgG1 and HER2-TRAIL control antibodies. Concomitantly, CD19^−^ CEM cells remained unaffected by CD19-TRAIL treatment (Figure 3C). Together, these data suggest that CD19-TRAIL kills BCP-ALL cells specifically by apoptosis induction.

### 3.3. CD19-TRAIL Eradicates BCP-ALL Cells In Vivo in Xenograft Models

Encouraged by the potent cell-toxic effects of CD19-TRAIL in vitro, we next investigated the therapeutic effect of CD19-TRAIL in vivo in BCP-ALL xenograft models. REH and NALM-6 cells were injected intravenously into NSG mice and animals treated with CD19-TRAIL on days +1, +3, +6, +10, +13 as published previously [19] and every 7 days thereafter or left untreated (*n* = 6/group). Animals were sacrificed when showing clinical signs of overt leukemia. Indeed, CD19-TRAIL treatment led to a clear prolongation of median mouse survival as compared to control animals in animals transplanted with REH cells (34 vs. 48 days, *p* = 0.0009, Figure 4A) and also in mice bearing NALM-6 cells (19.5 vs. 29 days, *p* = 0.0009, Figure 4B). Moreover, as expected, CD19-TRAIL outperformed CD19-IgG1.

We found a significant reduction in leukemia engraftment in bone marrow aspirates of animals bearing REH cells and in the peripheral blood of NALM-6-bearing mice that were treated with CD19-TRAIL as compared to animals treated with CD19-IgG1, respectively (38.30% vs. 17.70%, *p* = 0.0476 and 43.5% vs. 24%, *p* = 0.0079, Appendix A). In REH cells, bone marrow was analyzed because these cells do not cause leukemia in the peripheral blood of NSG mice.

As in vivo experiments using cell lines do not depict the clinical heterogeneity of BCP-ALL patients, we next tested CD19-TRAIL in patient-derived BCP-ALL xenografts (PDX) from four patients (Patients 1–4, Appendix A, *n* = 5 PDX mice/group). All patients exposed cell surface expression of CD19 and TRAIL-R1. One of four patients also exhibited weak TRAIL-R2 expression (Appendix A). Indeed, CD19-TRAIL treatment was efficient in all tested PDX-samples in vivo and significantly prolonged the survival of PDX-animals irrespective of ALL cytogenetics (Appendix A; Figure 4C–F). For patient 1 (BCR-ABL+), no CD19-TRAIL-treated animal showed clinical signs of leukemia up to day +160, whereas all control animals were sacrificed due to overt leukemia on day +80 (*p* = 0.0035). Furthermore, CD19-TRAIL-treated mice bearing PDX-cells of patient 2 (BCR-ABL+) showed a median survival prolongation of 35 days as compared to control animals (77 vs. 112 days, *p* = 0.0027). CD19-TRAIL-treated PDX-mice from patient 3 (E2A-PBX1) showed a median survival prolongation of 39 days (115 vs. 154 days, *p* = 0.0021). Moreover, PDX-mice of patient 4 (MLL-rearrangement) showed a significant median survival prolongation of 32 days upon CD19-TRAIL treatment as compared to control mice (58 vs. 90 days, *p* = 0.0023). Of note, 2 out 5 PDX-mice of patient 3 and 4 had not developed leukemia up to day +160, respectively (Figure 4E,F).

These results indicate that CD19-TRAIL efficiently targets CD19^+^ BCP-ALL cells in vivo in xenograft models.

### 3.4. The Cytotoxic Effect of CD19-TRAIL Is Synergistically Enhanced by Venetoclax in BCP-ALL Cells

Cancer drugs may initiate apoptosis through activation of the extrinsic or intrinsic pathways [54]. TRAIL acts as an activator of the extrinsic pathway of apoptosis by binding to death receptors [26,54]. On the other hand, Venetoclax (VTX) inhibits BCL-2, thereby inducing the release of the pro-apoptotic BH3-domain family members BAX and BIM, which act as key molecules of the intrinsic apoptosis pathway [54,55,56]. We hypothesized that the therapeutic efficacy of CD19-TRAIL could be further improved by dual stimulation of different apoptosis pathways. To test this hypothesis, REH cells were treated with CD19-TRAIL or VTX as monotherapies, or with the combination of both. Indeed, we detected a significant and dose-dependent increase in apoptosis induction by co-treatments with CD19-TRAIL and VTX as compared to single and control treatments in REH cells (Figure 5A). Of note, we found that the observed combinatorial pro-apoptotic effects of CD19-TRAIL and VTX were indeed synergistic as determined by synergy analysis according to Chou and Talalay [49] (Appendix A). To substantiate further these results, REH cells were treated with DMSO, CD19-TRAIL, VTX, or the combination, and cell lysates were analyzed by western blot. Protein levels of all cleaved apoptotic markers (Caspase 3, 7, 9, and PARP) as well as γH2AX, a marker of DNA-damage, were increased by co-treatment with CD19-TRAIL and VTX as compared to single and control treatments (Figure 5B). The synergy of CD19-TRAIL and VTX was then further investigated in vivo. NSG mice were injected with REH cells and treated with CD19-TRAIL, VTX, or CD19-TRAIL and VTX, as compared to untreated control animals (*n* = 6/group). Both CD19-TRAIL and VTX treatments as monotherapies significantly prolonged median survival of mice in comparison to untreated control animals by 14 days (34 vs. 48 days for both drugs, *p* = 0.0009, Figure 5C). Of note, a further prolongation of survival by 29 days was achieved by the combination of CD19-TRAIL and VTX as compared to control (34 vs. 63 days, respectively, *p* = 0.0009, Figure 5C). Survival prolongation by the combination was also significant compared to the respective monotherapies (48 vs. 63 days, *p* = 0.0009, respectively, Figure 5C). These data suggest that the efficacy of CD19-TRAIL is enhanced by a combination with the BCL-2 inhibitor VTX, which can be a novel combination strategy for clinical use.

## 4. Discussion

Antibody-based immunotherapy is an attractive tool to reduce the toxicity of chemotherapy and to induce target-cell-specific effects either in dependence of the patient’s immune system or by directly inducing anti-tumoral effects.

Due to its favorable expression profile, CD19 became a major target for immunotherapy approaches in BCP-ALL. Yet, novel CD19-directed antibodies such as the BiTE molecule blinatumomab harbor a significant toxicity profile due to the side effects of T-cell activation [11]. Native CD19 antibodies reveal a beneficial pharmacokinetic profile with long serum half-lives and low off-target toxicity, but a clinical pilot study using a mouse CD19-IgG2a antibody showed a limited response in B-cell non-Hodgkin’s lymphoma patients, probably due to limited effector functions elicited by the native CD19 antibody [16]. Therefore, the current study objective was to generate and characterize a novel CD19-directed antibody therapeutic based on a humanized CD19-IgG1 antibody and fused with the apoptosis-inducing ligand TRAIL, in order to induce BCP-ALL-specific killing. Our data demonstrate that a CD19-TRAIL fusion protein shows potent selective pro-apoptotic activity towards CD19-expressing cells in vitro. Furthermore, CD19-TRAIL significantly prolonged the survival of xenograft mice bearing human BCP-ALL cell lines and PDX cells, and its efficacy could be enhanced by the addition of Venetoclax.

We and others had previously reported that the efficacy of antibodies with low cytolytic capacity can be significantly enhanced by Fc-engineering: For example, the CD19 antibody tafasitamab (MOR208) showed increased FcγR-dependent NK-cell and macrophage recruitment than its wildtype counterpart and has been approved for clinical use [15,57]. Furthermore, we recently showed that a S267E/H268F/S324T/G236A/I332E (EFTAE) Fc-modification equipped a CD19 antibody with enhanced CDC-inducing features. This effect was further improved by reducing the fucose content, which enhanced the affinity of the antibody to FcγRIIIa and thereby increased ADCC [17].

A further option to increase antibody efficacy is to “arm” it with cytotoxic agents. The CD19-ADC loncastuximab tesirine and coltuximab ravtansine (SAR3419) exhibited strong anti-tumoral effects in hematological malignancies in vitro and an acceptable safety profile in patients but only modest efficacy in relapsed/refractory BCP-ALL [14,58].

A previous report showed lower off-target toxicity of a TRAIL antibody construct as compared to a pseudomonas exotoxin A (ETA)-linked counterpart, so that our results indicate that equipping the Fc region of antibodies with a TRAIL ligand may represent a further promising strategy to enhance their direct apoptosis-inducing abilities [59].

TRAIL is a highly potent apoptosis inducer that exists either as a type II membrane protein (memTRAIL) or as a cleaved, soluble protein (sTRAIL) [24,60]. The observations that sTRAIL attacks tumor cells efficiently and selectively and that it is at the same time is well tolerated in patients motivated numerous preclinical and clinical studies testing the effect of sTRAIL in different tumor entities, yet with limited success [30,32,33]. TRAIL-fusion proteins may help to overcome some of the limitations by enhancing the stability of the TRAIL ligand, reducing the formation of detrimental high-molecular weight aggregates as compared to unbound sTRAIL [59], and efficiently promoting the accumulation of TRAIL-R1 and TRAIL-R2 on the surface of target cells [39]. This way, the antibody-TRAIL construct induces the DISC-formation and apoptosis also in cells that lack TRAIL-R1 which responds to sTRAIL and memTRAIL, whereas TRAIL-R2 preferentially responds to memTRAIL [40]. This is particularly valid for the scTRAIL containing three TRAIL protomers of the CD19-construct applied in our study [44]. Accordingly, we found that CD19-TRAIL induced apoptosis in NALM-6 cells, for which we detected no TRAIL-R1 expression. Moreover, CD19-TRAIL induced apoptosis in CD19^+^ cell lines REH and NALM-6 at nanomolar concentrations. Accordingly, we detected no induction of apoptosis by CD19-TRAIL in CD19^−^ cells or when using a HER2-TRAIL control antibody, indicating negligible binding of soluble antibody constructs to TRAIL-R1 and TRAIL-R2 via the TRAIL domain. We suspect that CD19-TRAIL potentiates the approximation and oligomerization of TRAIL-R-complexes which strongly enhances the efficiency of TRAIL-R-mediated apoptosis induction and that assembly of multiple CD19-TRAIL-R complexes is needed to induce apoptosis at particularly low concentrations. This is further substantiated by the observation that cell death was arrested after blocking the CD19 epitope with the parental CD19-IgG1 antibody, hampering the crosslinking between the TRAIL-Rs and the CD19 antigen. This is in line with previous studies showing target-specific eradication of tumors cells by TRAIL fusion proteins with a single chain variable fragment (scFv) [37,39,59]. Moreover, antibody-scTRAIL constructs were already shown to spare physiological blood cells from apoptosis induction [37].

The CD19-dependent cytotoxic activities of CD19-TRAIL may be supported by the Fc-modifications that were introduced to promote heterodimeric assembly of the HCs [43] and to generate an antibody carrying only one scTRAIL moiety. In its design, CD19-TRAIL is different to previously reported antibody-scTRAIL constructs, in which scTRAIL was linked to each LC or HC or both, resulting in hexavalent or dodecavalent scTRAIL constructs, respectively [20]. In our design, TRAIL-R activation in the absence of the CD19 antigen may be reduced further, which may result in superior antigen specificity as compared to multivalent TRAIL constructs.

CD19-TRAIL single treatment was efficient in reducing leukemic burden of BCP-ALL PDX cells from different patients in xenografted NSG-mice and was well tolerated by treated animals, which to our knowledge has not been reported for other CD19 antibody-TRAIL constructs to date. High preclinical efficacy in ALL-PDX-specimen has been reported for TRAIL fused to a CD19-ligand [61]. Comparing different TRAIL-based CD19-targeted therapies would be of high interest on the way to potential clinical application. Yet, the tolerability of CD19-TRAIL constructs would have to be validated in non-human primate models due to the minor interaction of human TRAIL ligands with murine TRAIL-R [62]. Moreover, such models could be used to investigate side effects such as aplasia of normal B cells, which is frequently observed for other CD19-targeted immunotherapies such as blinatumomab and CD19-specific chimeric antigen receptor T-cells [63,64] or cytotoxicity towards other normal blood cells.

Another interesting observation in our study is that the efficacy of CD19-TRAIL was significantly and synergistically enhanced when combined with the BCL-2 inhibitor VTX. VTX has shown promising effects and tolerability in different hematological malignancies including BCP-ALL [50,65]. Whereas CD19-TRAIL induces extrinsic apoptosis via DISC induction, VTX stimulates the intrinsic apoptosis pathway via release of pro-apoptotic BH3-domain family members BAX and BIM [26,54,56]. The concomitant stimulation of both apoptotic pathways may potentiate the efficacy of TRAIL-carrying monoclonal antibodies and may be of general interest for antibody-based immunotherapy.

CD19 loss remains a major challenge in BCP-ALL relapse and a key mechanism of the failure of CD19-directed therapy [10,66]. Our CD19-TRAIL construct may help to overcome this challenge by binding CD19^+^ cells and concomitantly crosslinking TRAIL receptors on neighboring tumor cells, thereby inducing apoptosis in these cells irrespective of CD19 expression (“bystander killing”) [67]. Moreover, the concept of sTRAIL fusion may also be applicable to antibodies targeting other important antigens in hematological malignancies such as CD20 and CD38 for which TRAIL fusion constructs were already described [68,69] or further novel targets such as CD52 and CD79 [53,70]. Furthermore, first antibody-based immunotherapy concepts targeting T-ALL such as daratumumab have shown promising preclinical efficacy [52], which could be enhanced by using TRAIL fusions. To this end, a scFvCD7-TRAIL antibody was already shown to induce T-ALL-specific killing, promoting the view that sTRAIL antibody constructs could also be an interesting option to target T-ALL cells [59].

Taken together, the preclinical data presented here suggest that CD19-TRAIL may represent a promising new tumor cell-specific therapeutic agent in BCP-ALL, warranting further preclinical testing.

## Figures and Tables

**Figure 1 jcm-10-02634-f001:**
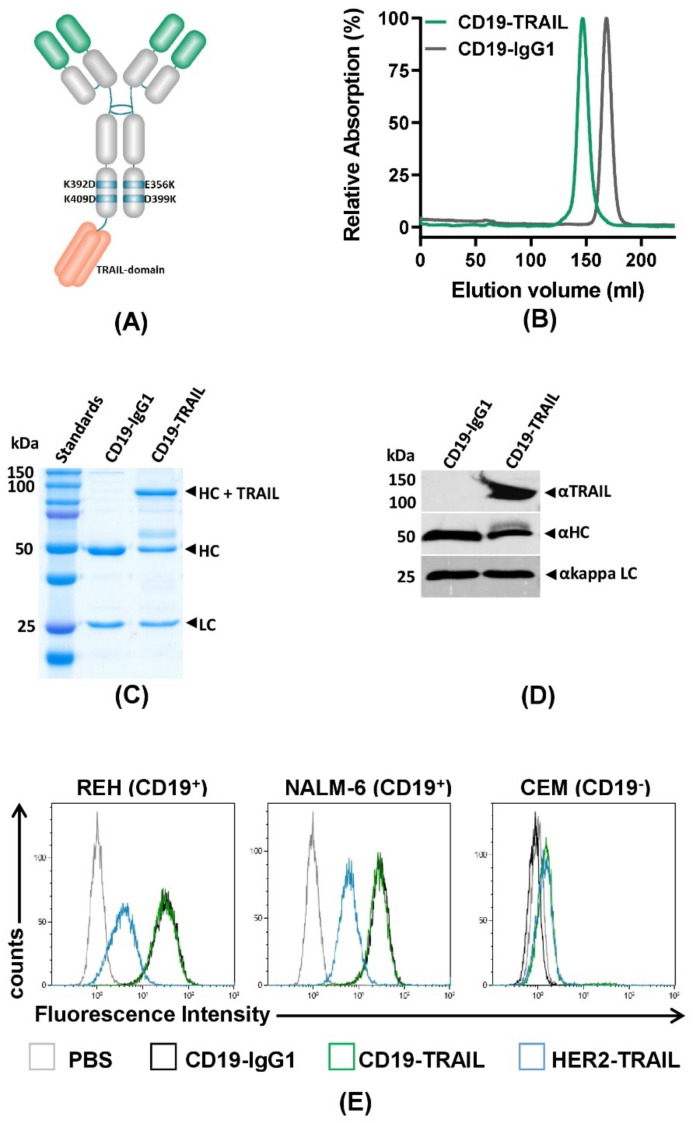
Generation of a target-specific CD19-TRAIL fusion construct: (**A**) Schematic illustration of a CD19-TRAIL fusion construct (CD19-TRAIL) consisting of one heavy chain with two amino acid exchanges (E356K and D399K) and a second heavy chain carrying two amino acid exchanges (K392D and K409D) and a C-terminal fusion of a single chain TRAIL domain, resulting in a heterodimeric antibody promoting TRAIL-receptor trimerization in CD19-positive (CD19^+^) target cells; (**B**) Purity of CD19-TRAIL and CD19-IgG1 was analyzed by size exclusion chromatography under native buffer conditions, and peak fractions were collected. Representative chromatography image of the isolated and re-analyzed peak fraction (normalized to maximum absorption) is shown; (**C**,**D**) Purity and molecular masses of CD19-TRAIL and CD19-IgG1 were further validated using SDS-PAGE under reducing conditions followed by staining with Coomassie blue and D) Western blot analyses with immunodetection of TRAIL as well as antibody heavy and light chains. HC = heavy chain, LC = light chain; (**E**) Specificity of the binding capacity of CD19-TRAIL and CD19-IgG1 compared to a HER2-TRAIL control antibody was tested on the CD19^+^ BCP-ALL cell lines REH and NALM-6 cells and the CD19-negative (CD19^−^) T-ALL cell line CEM by flow cytometry analyses. Antibodies were detected with a FITC-conjugated anti-human IgG Fc F(ab′)2 secondary antibody. PBS served as control for background staining. Depicted data show representative pictures of *n* = 3 experiments.

**Figure 2 jcm-10-02634-f002:**
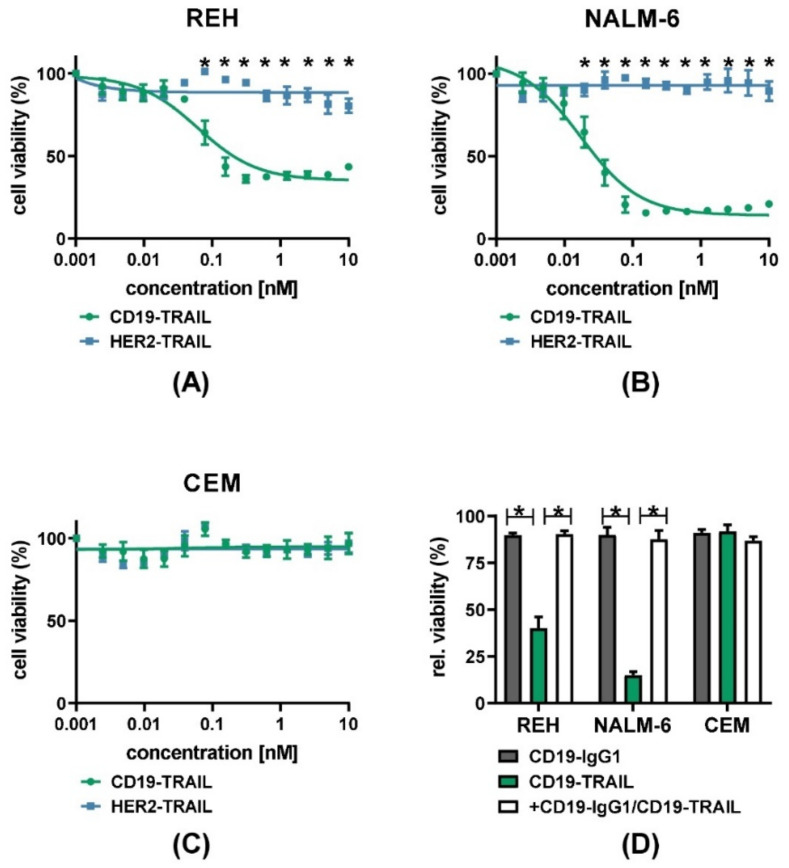
CD19-TRAIL elicits selective anti-proliferative effects in CD19^+^ ALL-cells: (**A**) CD19^+^ REH and (**B**) NALM-6 cells as well as (**C**) CD19^−^ CEM cells were treated with escalating concentrations of CD19-TRAIL or HER2-TRAIL for 72 h, and cell viability was analyzed by MTT assay, two-way analysis of variance; (**D**) Specificity of CD19-dependant cell killing was verified by pre-incubating the cell lines REH, NALM-6, and CEM with CD19-IgG1 antibody (100 nM) for 1 h prior to CD19-TRAIL treatment and measuring cell viability by MTT assay after 72 h, one-tailed Mann–Whitney Test. Graphs depict mean values ± SEM of *n* = 3 independent experiments. * *p* < 0.05.

**Figure 3 jcm-10-02634-f003:**
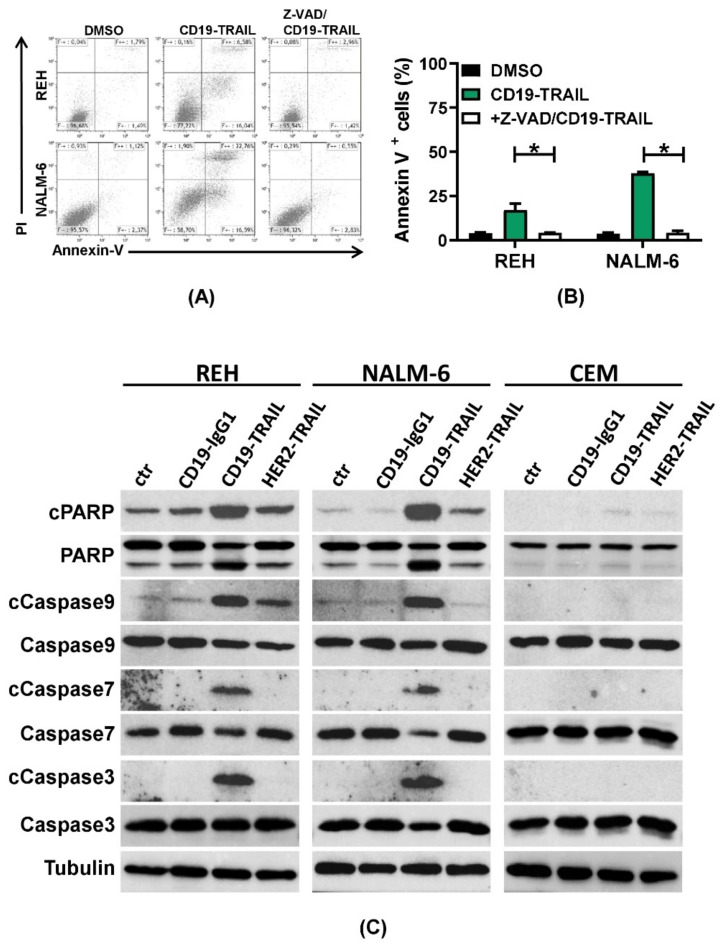
CD19-TRAIL kills ALL-cells via apoptosis induction: (**A**,**B**) REH and NALM-6 cells were pre-incubated with the pan-caspase inhibitor Z-VAD (50 µM) for 1 h prior to incubation with CD19-TRAIL. DMSO served as negative control; (**A**) Representative histograms of flow cytometric analyses of AnnexinV and PI staining after 24 h and (**B**) mean values ± SEM of *n* = 3 independent experiments are shown, one-tailed Mann–Whitney Test, * *p* < 0.05; (**C**) Representative images of western blot analyses of the pro-apoptotic proteins Caspase 3, Caspase 7, and Caspase 9 and PARP as well as their cleavage (c) products (cCaspase3, cCaspase7, cCaspase9, and cPARP) in the cell lines REH, NALM-6, and CEM after treatment with 0.5 nM of CD19-IgG1, CD19-TRAIL, HER2-TRAIL, or medium (ctr) for 48 h. Tubulin served as loading control.

**Figure 4 jcm-10-02634-f004:**
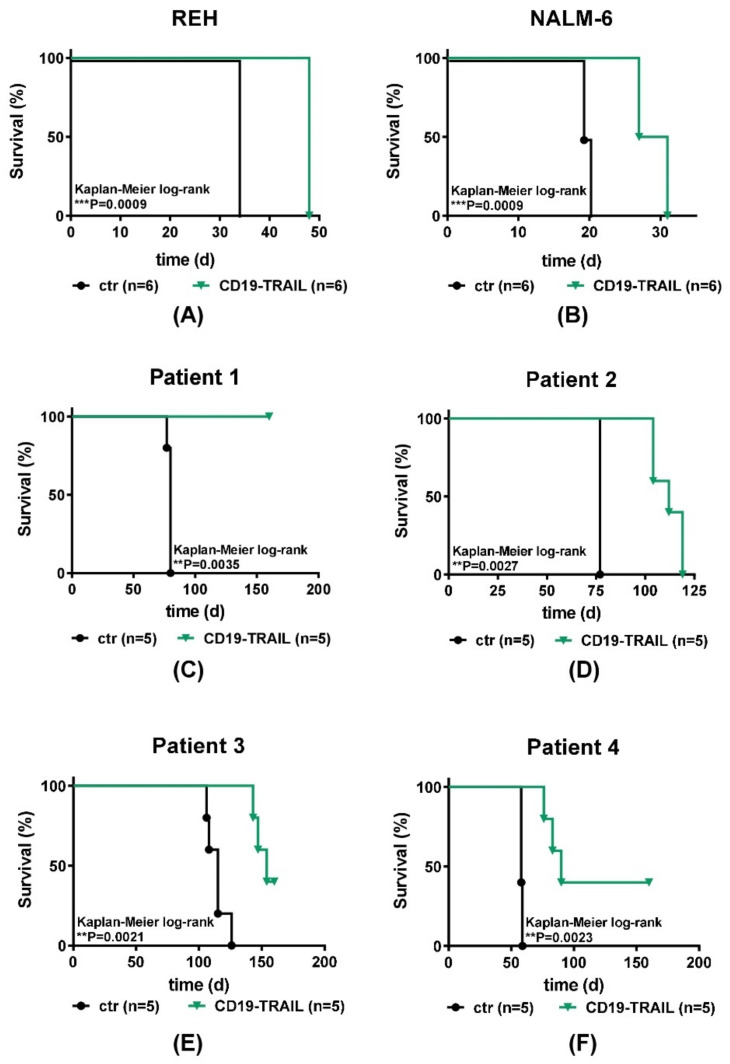
CD19-TRAIL eradicates BCP-ALL cells in vivo. Survival of female NSG-mice intravenously injected with (**A**) 0.5 × 10^6^ REH (*n* = 6 per group); (**B**) 0.5 × 10^6^ NALM-6 (*n* = 6 per group); or (**C**–**F**) 1 × 10^5^ patient-derived ALL xenograft (PDX; *n* = 5 per group) cells (day 0). On days +1, +3, +6, +10, +13 and every 7 days thereafter, mice were treated with 1.5 mg/kg of CD19-TRAIL intravenously or left untreated (ctr). At signs of overt leukemia (detection of >75% leukemic blasts in the peripheral blood or clinical signs of leukemia including loss of weight or activity, organomegaly, hind-limb paralysis), mice were euthanized. Differences in survival were calculated using Kaplan–Meier log-rank test, ** *p* < 0.01, *** *p* < 0.001.

**Figure 5 jcm-10-02634-f005:**
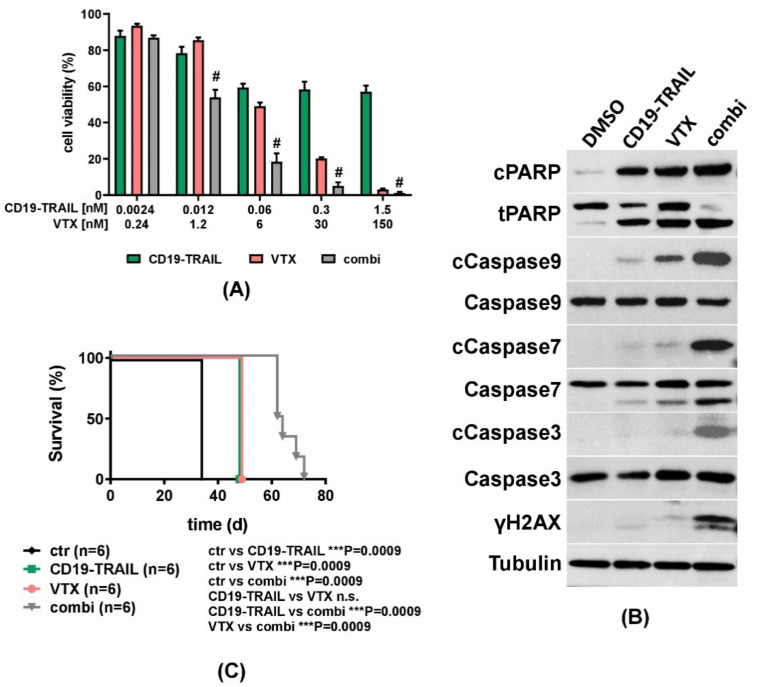
ALL-killing effect of CD19-TRAIL is synergistically enhanced by Venetoclax: (**A**) REH cells were treated with escalating concentrations of CD19-TRAIL, the BCL-2 inhibitor Venetoclax (VTX), or the combination of both (combi). Cell viability was determined by MTT assay after 72 h. Combination Index (CI) was calculated via CompuSyn Software, and synergism (CI < 1) is indicated by (#). Values for CI calculations are noted in Appendix A. Data represent mean values ± SEM of *n* = 3 independent experiments; (**B**) Representative western blot analyses of REH cells treated with DMSO, CD19-TRAIL (0.06 nM), VTX (6 nM), or the combination (combi) for 48 h. Expression levels of indicated proteins were determined in whole-cell protein extracts. Tubulin served as a loading control; (**C**) Female NSG-mice were intravenously injected with 0.5 × 10^6^ REH cells (*n* = 6 per group) and treated with 1.5 mg/kg of CD19-TRAIL intravenously on days +1, +3, +6, +10, +13 and every 7 days thereafter; 100 mg/kg VTX daily oral gavage; or both (combi). Untreated mice served as control (ctr). When showing signs of overt leukemia (clinical signs of leukemia including loss of weight or activity, organomegaly, hind-limb paralysis), mice were euthanized, and differences in mouse survival were calculated using Kaplan–Meier log-rank test, *** *p* < 0.001.

## Data Availability

The data presented in this study are available in the main manuscript or Appendix A.

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
