# Peer review of "Engineering of CD19 Antibodies: A CD19-TRAIL Fusion Construct Specifically Induces Apoptosis in B-Cell Precursor Acute Lymphoblastic Leukemia (BCP-ALL) Cells In Vivo"

_jcm, 2021, doi:10.3390/jcm10122634_

Round 1

Reviewer 1 Report

  1. Please provide brief rationale for use of your mouse strain
  2. Please provide reference for clinical signs of leukemia in mice
  3. Figure 1 E shows the fusion construct can bind via TRAILR. Please briefly clarify why this does not, by itself, cause cell death. 
  4. Clarify why different time points were used for different apoptosis-related assays (reference a time course experiment, for example. Not necessary to show). 
  5. There is disconnect between induction of apoptosis via TRAILR and the apoptotic proteins assayed. Was Caspase 8 activated/cleaved in CD19-TRIAL-treated B-ALL cells? Can you show DISC assembly? This needs to be addressed. A diagram outlining the apoptotic pathway based on your data might be helpful. 
  6. Figure 3C shows cleavage products of caspases 3, 7, and 9 in REH cells after 24h exposure to CD19-TRAIL. However, Figure 5B shows minimal caspase 9 cleavage and no obvious cleavage of casepases 3 and 7. Please address. 
  7. Supplementary figure 3, description in the figure legend refers to CD19-IgG1 or CD19-TRAIL blocking the CD19 epitope on NALM-6 cells, preventing flow assessment of blasts in the mice. Why, then, can CD19+ REH blasts be detected by flow? Please address. 

Reviewer 2 Report

Winterberg et al, is a well-written paper describing generation of a fusion CD19-TRAIL protein, which, when presented on the surface of the malignant human hematopoietic B-cells, causes apoptosis and sensitize them to chemotherapeutic drugs. I would not have hesitations in recommending this paper for publication, and only have one question to authors:

Did you try CD19-TRAIL antibodies on the umbilical CD34+ cord blood cells or patient-specific common lymphoid and myeloid progenitors to demonstrate that CD19-TRAIL has less or no cytotoxicity to normal human blood cells?

Author Response

Reviewer #2:

Winterberg et al, is a well-written paper describing generation of a fusion CD19-TRAIL protein, which, when presented on the surface of the malignant human hematopoietic B-cells, causes apoptosis and sensitize them to chemotherapeutic drugs. I would not have hesitations in recommending this paper for publication, and only have one question to authors:

Did you try CD19-TRAIL antibodies on the umbilical CD34+ cord blood cells or patient-specific common lymphoid and myeloid progenitors to demonstrate that CD19-TRAIL has less or no cytotoxicity to normal human blood cells?

Response: We thank the reviewer for the positive view of our manuscript and the interesting question. We have not investigated the impact on CD19-TRAIL on physiological blood cells of individual patients in detail. Previous reports showed that (CD19)-TRAIL fusion constructs have no or only little impact on normal blood cells as shown by Uckun et al. 2015 and Bremer et al. 2004. Yet, we cannot rule out that treatment with CD19-TRAIL may lead to side effects like B-cell aplasia which is frequently observed for other CD19-targeted immunotherapeutic approaches such as blinatumomab and CD19-specific chimeric antigen receptor T-cells treatment (Maude et al. 2014; Queudeville et al. 2021). We added a sentence to the discussion section of the manuscript acknowledging the need to further investigate this important issue.

References

Bremer, Edwin; Kuijlen, Jos; Samplonius, Douwe; Walczak, Henning; Leij, Lou de; Helfrich, Wijnand (2004): Target cell-restricted and -enhanced apoptosis induction by a scFv:sTRAIL fusion protein with specificity for the pancarcinoma-associated antigen EGP2. In International journal of cancer 109 (2), pp. 281–290. DOI: 10.1002/ijc.11702.

Chou, Ting-Chao; Talalay, Paul (1984): Quantitative analysis of dose-effect relationships: the combined effects of multiple drugs or enzyme inhibitors. In Advances in Enzyme Regulation 22, pp. 27–55. DOI: 10.1016/0065-2571(84)90007-4.

Maude, Shannon L.; Frey, Noelle; Shaw, Pamela A.; Aplenc, Richard; Barrett, David M.; Bunin, Nancy J. et al. (2014): Chimeric antigen receptor T cells for sustained remissions in leukemia. In The New England journal of medicine 371 (16), pp. 1507–1517. DOI: 10.1056/NEJMoa1407222.

Queudeville, Manon; Schlegel, Patrick; Heinz, Amadeus T.; Lenz, Teresa; Döring, Michaela; Holzer, Ursula et al. (2021): Blinatumomab in pediatric patients with relapsed/refractory B-cell precursor acute lymphoblastic leukemia. In European journal of haematology 106 (4), pp. 473–483. DOI: 10.1111/ejh.13569.

Schewe, Denis M.; Alsadeq, Ameera; Sattler, Cornelia; Lenk, Lennart; Vogiatzi, Fotini; Cario, Gunnar et al. (2017): An Fc-engineered CD19 antibody eradicates MRD in patient-derived MLL-rearranged acute lymphoblastic leukemia xenografts. In Blood 130 (13), pp. 1543–1552. DOI: 10.1182/blood-2017-01-764316.

Townsend, E. C.; Murakami, M. A.; Christodoulou, A.; Christie, A. L.; Köster, J.; DeSouza, T. A. et al. (2016): The Public Repository of Xenografts Enables Discovery and Randomized Phase II-like Trials in Mice. In Cancer cell 29 (4). DOI: 10.1016/j.ccell.2016.03.008.

Uckun, Fatih M.; Myers, Dorothea E.; Qazi, Sanjive; Ozer, Zahide; Rose, Rebecca; D'Cruz, Osmond J.; Ma, Hong (2015): Recombinant human CD19L-sTRAIL effectively targets B cell precursor acute lymphoblastic leukemia. In The Journal of clinical investigation 125 (3), pp. 1006–1018. DOI: 10.1172/JCI76610.

Vogiatzi, Fotini; Winterberg, Dorothee; Lenk, Lennart; Buchmann, Swantje; Cario, Gunnar; Schrappe, Martin et al. (2019): Daratumumab eradicates minimal residual disease in a preclinical model of pediatric T-cell acute lymphoblastic leukemia. In Blood 134 (8), pp. 713–716. DOI: 10.1182/blood.2019000904.

Walsh, Nicole; Kenney, Laurie; Jangalwe, Sonal; Aryee, Ken-Edwin; Greiner, Dale L.; Brehm, Michael A.; Shultz, Leonard D. (2017): Humanized mouse models of clinical disease. In Annual review of pathology 12, pp. 187–215. DOI: 10.1146/annurev-pathol-052016-100332.

Round 2

Reviewer 1 Report

I thank the authors of this manuscript for addressing my comments.